# D³Fields: Dynamic 3D Descriptor Fields for Zero-Shot Generalizable Robotic Manipulation

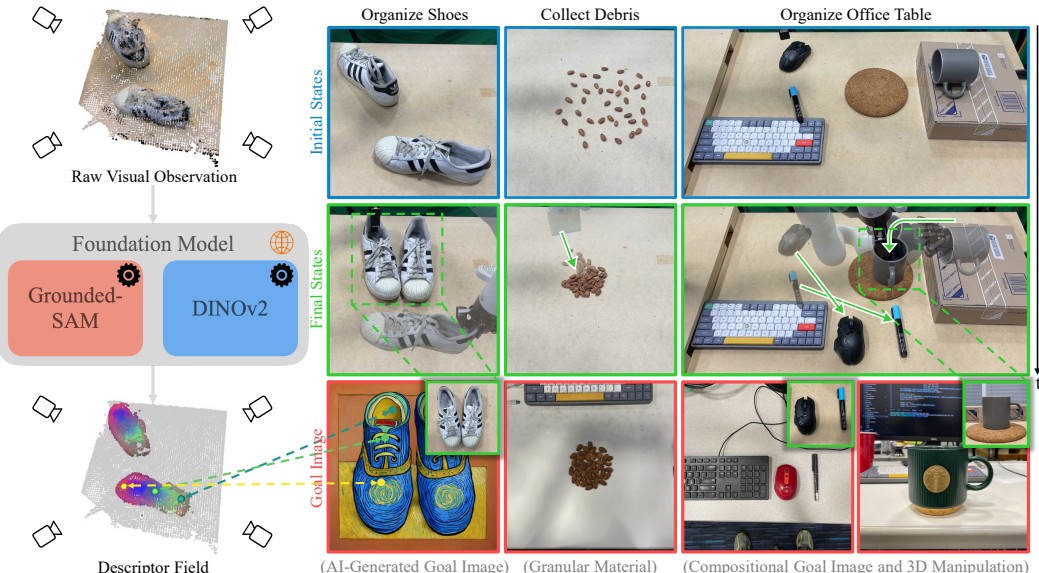

Figure 1: **D³Fields Representation and Application to Various Manipulation Tasks.** D³Fields take in multi-view RGBD images and encode semantic features and instance masks using foundational models. The gray and colored points in the bottom left visualize background and semantic features mapped to RGB space using Principal Component Analysis (PCA), demonstrating consistency across instances. We use our representation for diverse tasks in a zero-shot manner. These tasks are defined by 2D goal images with diverse instances and styles. We address pick-and-place tasks such as shoe organization and tasks requiring dynamic modeling like collecting debris. We also demonstrate in the office table organization that our framework can accomplish 3D manipulation and compositional task specification.

**Abstract:**

Scene representation has been a crucial design choice in robotic manipulation systems. An ideal representation should be 3D, dynamic, and semantic to meet the demands of diverse manipulation tasks. However, previous works often lack all three properties simultaneously. In this work, we introduce D³Fields — dynamic 3D descriptor fields. These fields capture the dynamics of the underlying 3D environment and encode both semantic features and instance masks. Specifically, we project arbitrary 3D points in the workspace onto multi-view 2D visual observations and interpolate features derived from foundational models. The resulting fused descriptor fields allow for flexible goal specifications using 2D images with varied contexts, styles, and instances. To evaluate the effectiveness of these descriptor fields, we apply our representation to a wide range of robotic manipulation tasks in a zero-shot manner. Through extensive evaluation in both real-world scenarios and simulations, we demonstrate that D³Fields are both generalizable and effective for zero-shot robotic manipulation tasks. In quantitative comparisons with state-of-the-art dense descriptors, such as Dense Object Nets and DINO, D³Fields exhibit significantly better generalization abilities and manipulation accuracy.

Submitted to the 7th Conference on Robot Learning (CoRL 2023). Do not distribute.

# 1 Introduction

The choice of scene representation is critical in robotic systems. An ideal representation should be simultaneously 3D, dynamic, and semantic to meet the needs of various robotic manipulation tasks in our daily lives. However, previous research into scene representations in robotics often does not encompass all three properties. Some representations exist in 3D space [1, 2, 3, 4], yet they overlook semantic information. Others focus on dynamic modeling [5, 6, 7, 8], but only consider 2D data. Some other works are limited by only considering semantic information such as object instance and category [9, 10, 11, 12, 13].

In this work, we aim to satisfy all three criteria by introducing $D^3$Fields, unified descriptor fields that are 3D, dynamic, and semantic. $D^3$Fields take in arbitrary points in the 3D world coordinate frame and output both geometric and semantic information related to these points. This includes the instance mask, dense semantic features, and the signed distance to the object surface. Notably, deriving these descriptor fields requires no training and is conducted in a zero-shot manner using large foundational vision models and vision-language models (VLMs). Specifically, we first use Grounding-DINO [14], Segment Anything (SAM) [15], XMem [16], and DINOv2 [17] to extract information from multi-view 2D RGB images. We then project the 3D points back to each camera, interpolate to compute representations from each view, and fuse these data to derive the descriptors for the associated 3D points, as shown in Fig. 1 (left). By leveraging the dense semantic feature and instance mask of our representation, we can robustly track 3D points of the target object instance and train dynamics models. These learned dynamics models can then be incorporated into a Model-Predictive Control (MPC) framework to plan for manipulation tasks.

Notably, the derived representations allow for goal specification using 2D images sourced from the Internet, phones, or those generated by AI models. Such goal images have been challenging to manage with previous methods, because they contain varied styles, contexts, and object instances different from the robot's workspace. Our proposed $D^3$Fields can establish dense correspondences between the robot workspace and the target configurations. These correspondences give us the task objective, enabling us to plan the robot's actions with the learned dynamics model within the MPC framework. This task execution process does not require any further training, offering a flexible and convenient interface for humans to instruct robots.

We evaluate our method across a wide range of household robotic manipulation tasks in a zero-shot manner. These tasks include organizing shoes, collecting debris, and organizing office desks, as shown in Fig. 1 (right). Furthermore, we offer detailed quantitative comparisons between our method and other state-of-the-art dense descriptor techniques. Our results indicate that our approach significantly outperforms in terms of generalizability and manipulation accuracy.

To summarize our contributions: (1) We introduce a novel representation, $D^3$Fields, that is **3D**, **dynamic**, and **semantic**. (2) We present a novel and flexible goal specification method using 2D images that incorporate a range of styles, contexts, and instances. (3) Our proposed robotic manipulation framework supports zero-shot generalizable manipulation applicable to a broad spectrum of household tasks.

# 2 Related Works

## 2.1 Foundation Models for Robotics

Foundation models generally refer to those trained on broad data, often using self-supervision at scale, which can then be adapted (e.g., fine-tuned) to various downstream tasks. Large Language Models (LLMs) have showcased promising reasoning abilities for language. Robotics researchers have recently released a series of works that leverage LLMs, including SayCan [18] and Inner Mono-logue [19], to directly generate robot plans. Some later works have used LLMs as a code generator: Code as Policies [20] uses 2D object detectors as the perception API, whereas VoxPoser [21] creates a 3D value map. Yet, their perception modules fall short in modeling the precise geometry and

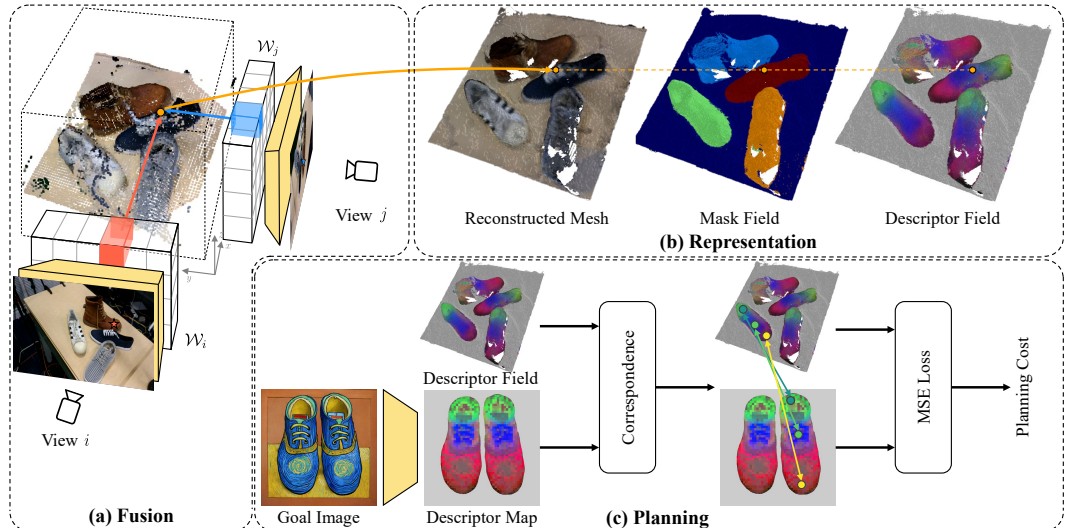

Figure 2: **Overview of the proposed framework.** (a) The fusion process fuses RGBD observations from multiple views. Each view is processed by foundation models to obtain the feature volume $\mathcal{W}$. Arbitrary 3D points are processed through projection and interpolation. (b) After fusing information from multiple views, we obtain an implicit distance function to reconstruct the mesh form. We also have instance masks and semantic features for evaluated 3D points, as shown by the mask field and descriptor field in the top right subfigure. (c) Given a 2D goal image, we use foundation models to extract the descriptor map. Then we correspond 3D features to 2D features and define the planning cost based on the correspondence.

dynamics of objects. Our D³Fields aim to address this by focusing on detailed 3D geometry and dynamics.

Meanwhile, foundational vision models, such as SAM [15] and DINOv2 [17], have demonstrated impressive zero-shot generalization capabilities across various vision tasks. However, their focus is primarily on 2D vision tasks. Grounding these models in a dynamic 3D environment remains a challenge. The recent GROOT project showcases how to construct 3D object-centric representations using foundational models and exhibits notable few-shot generalization capabilities [22]. Still, GROOT does not emphasize learning about object dynamics or achieving zero-shot generalizable robotic manipulation.

## 2.2 Representation for Visual Robotic Manipulation

Scene representation has been a pivotal component in robotic manipulation systems. Some early work relies on 2D representations, such as bounding boxes [23, 24]. Many recent methods construct particle representations of the environment and employ learned dynamics to capture the system's underlying structure [25, 3, 7, 8, 26, 27, 28, 29]. They demonstrate impressive results in unstructured environments and with non-rigid objects. However, they are not semantic, which can hinder their ability to generalize to new tasks and scenarios. Some research opts for a fixed-dimension latent vector derived from high-dimensional sensory inputs as the representation [30, 5, 6, 31, 32, 33, 34, 35, 36, 2], but such a representation does not scale well to complex manipulation tasks that require high precision and explicit scene structures. Other approaches use 6 DoF object poses as their representation [9, 10, 37, 38], though focusing primarily on grasping tasks instead of more dynamic ones. In this work, we aim to address these issues by introducing D³Fields, a representation that models dynamic 3D environments at varying semantic levels.

## 2.3 Neural Fields for Robotic Manipulation

Researchers have presented a variety of works using neural fields as a representation for robotic manipulation [39, 40, 41, 42, 43, 44, 45, 46, 47, 48, 49, 50, 51, 52, 41]. Among them, Neural Descriptor Fields are the most relevant to ours [42]. They build neural feature fields that generalize to

different instances with several demonstrations; but they focus on learning geometric, not semantic features, which hinders cross-category generalization.

Recently, a series of works distilled neural feature fields using foundation models such as CLIP and DINO for supervision [53, 54]. LeRF distills neural feature fields to handle open-vocabulary 3D queries and develops task-oriented grasping based on it [55, 56]. Shen et al. [57] use a similar distilled feature field for the grasping task. Both methods require dense camera views to train the neural field. GNFactor addresses this by introducing a voxel encoder [58]. However, distilling foundation models to create neural feature fields has drawbacks: (1) They often require dense camera views for a quality field. (2) Distilled neural fields need retraining for new scenes, limiting their generalization and making them ineffective for dynamic scenes. In contrast, our $D^3$Fields do not need extra training for new scenes and can work with sparse views and dynamic settings.

## 3 Method

In this section, we introduce the problem formulation in Section 3.1 and define camera transformation and projection notations in Section 3.2. The construction of $D^3$Fields is detailed in Section 3.3. Section 3.4 discusses tracking keypoints and learning dynamics, while Section 4.3 showcases how our representation enables zero-shot generalizable manipulation skills.

### 3.1 Problem Formulation

Given a 2D goal image $\mathcal{I}$, we denote the corresponding scene representation as $s_{\text{goal}}$. Our goal is to find the action sequence $\{a^t\}$ to minimize the task objective:

$$\min_{\{a_t\}} \quad c(s^T, s_{\text{goal}}), \\ \text{s.t.} \quad s^t = g(o^t), \quad s^{t+1} = f(s^t, a^t), \tag{1}$$

where $c(\cdot, \cdot)$ is the cost function measuring the distance between the terminal representation $s^T$ and the goal representation $s_{\text{goal}}$. Representation extraction function $g(\cdot)$ takes in the current multi-view RGBD observations $o^t$ and outputs the current representation $s^t$. $f(\cdot, \cdot)$ is the dynamics function that predicts the future representation $s^{t+1}$, conditioned on the current representation $s^t$ and action $a^t$. The optimization aims to find the action sequence $\{a_t\}$ that minimizes the cost function $c(s^T, s_{\text{goal}})$.

### 3.2 Notation: Camera Transformation and Projection

We assume all cameras' intrinsic parameters $\mathbf{K}$ and extrinsic parameters $\mathbf{T}$ are known. The camera $i$ extrinsic parameters are defined as follows.

$$\mathbf{T}_i = \begin{bmatrix} \mathbf{R}_i & \mathbf{t}_i \\ 0^T & 1 \end{bmatrix} \in \mathbb{SE}(3), \tag{2}$$

where Euclidean group $\mathbb{SE}(3) := \{\mathbf{R}, \mathbf{t} \mid \mathbf{R} \in \mathbb{SO}^3, \mathbf{t} \in \mathbb{R}^3\}$. For a 3D point $\mathbf{x}$ in the world frame, we could obtain projected pixel $\mathbf{u}_i$ and distance to camera $\mathbf{r}_i$ as follows:

$$\mathbf{u}_i = \pi\left(\mathbf{K}_i\left(\mathbf{R}_i\mathbf{x} + \mathbf{t}_i\right)\right), \quad \mathbf{r}_i = [0, 0, 1]^T\left(\mathbf{R}_i\mathbf{x} + \mathbf{t}_i\right), \tag{3}$$

where $\pi$ performs perspective projection, mapping a 3D vector $p = [x, y, z]^T$ to a 2D vector $q = [x/z, y/z]^T$.

### 3.3 $D^3$Fields Representation

We fuse observation $o^t$ from multiple views to build the implicit 3D descriptor fields $\mathcal{F}^t(\cdot)$. For simplicity, we will represent $o^t$ as $o$, and $\mathcal{F}^t(\cdot)$ as $\mathcal{F}(\cdot)$ in this subsection. The implicit 3D descriptor field $\mathcal{F}(\cdot)$ is defined as

$$(\mathbf{d}, \mathbf{f}, \mathbf{p}) = \mathcal{F}(\mathbf{x}), \tag{4}$$

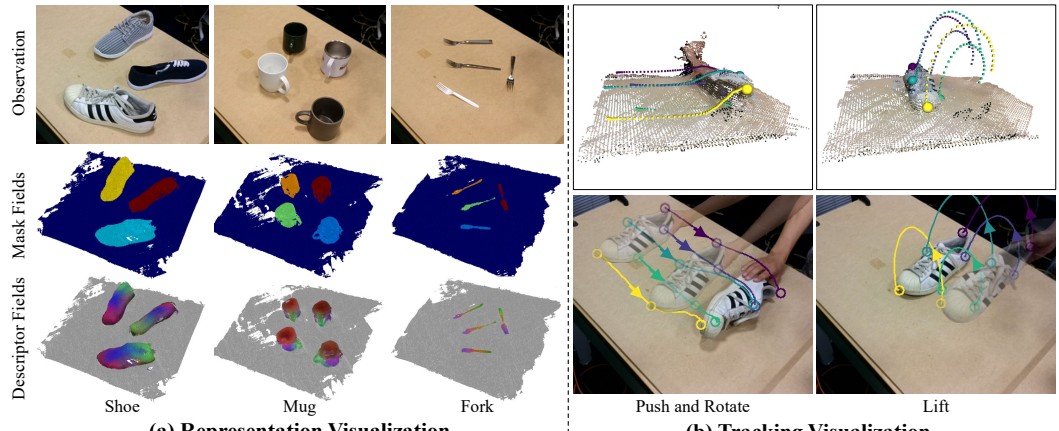

**(a) Representation Visualization**     **(b) Tracking Visualization**

Figure 3: **Representation and Tracking Visualizations.** (a) To verify that the representation is both 3D and semantic, we visualize the representation across different object categories. Mask fields color 3D points based on their instance masks, which clearly differentiates between instances. Descriptor fields color 3D points by mapping features to RGB space using PCA. They display a consistent color pattern within a category, such as mug handles being colorized as green for different mug instances. (b) To demonstrate that our representation is dynamic, we apply it to tracking tasks and showcase two tracking examples, both of which involve 3D motions and partial observations in single views. The robust 3D tracking results confirm that our representation is 3D, dynamic, and semantic.

where $\mathbf{x}$ is an arbitrary 3D point in the world frame, and $(\mathbf{d}, \mathbf{f}, \mathbf{p})$ is the corresponding geometric and semantic descriptor. $\mathbf{d} \in \mathbb{R}$ is the signed distance from $\mathbf{x}$ to the surface. $\mathbf{f} \in \mathbb{R}^N$ represents the semantic information of $N$ dimension. $\mathbf{p} \in \mathbb{R}^M$ denotes the instance probability distribution of $M$ instances. $M$ could be different across scenarios.

More specifically, we denote a single view RGBD observation from camera $i$ as $\boldsymbol{o}_i = (\mathcal{I}_i, \mathcal{R}_i)$, where RGB image $\mathcal{I}_i \in \mathbb{R}^{H \times W \times 3}$, and depth image $\mathcal{R}_i \in \mathbb{R}^{H \times W}$. For an arbitrary 3D point $\mathbf{x}$, we project it to image space using Eq. 3 and use bilinear interpolation to obtain the corresponding depth $\mathbf{r}'_i = \mathcal{R}_i[\mathbf{u}_i]$. Then the descriptors from camera $i$ are

$$\mathbf{d}_i = \mathbf{r}'_i - \mathbf{r}_i, \quad \mathbf{d}'_i = \max(\min(\mathbf{d}_i, \mu), -\mu),$$
$$\mathbf{f}_i = \mathcal{W}^{\mathbf{f}}_i[\mathbf{u}_i], \quad \mathbf{p}_i = \mathcal{W}^{\mathbf{P}}_i[\mathbf{u}_i], \tag{5}$$

where DINOv2 [17] extracts the semantic feature volume $\mathcal{W}^{\mathbf{f}}_i \in \mathbb{R}^{H \times W \times N}$ from RGB observation $\mathcal{I}_i$. $\mathcal{W}^{\mathbf{P}}_i \in \mathbb{R}^{H \times W \times M}$ is the instance mask volume using Grounded-SAM [14, 15]. $\mu$ is the truncation threshold for TSDF.

We fuse descriptors from all $K$ views as follows:

$$v_i = H(\mathbf{d}_i + \mu), \quad w_i = \exp\left(\frac{\min(\mu - |\mathbf{d}_i|, 0)}{\mu}\right), \tag{6}$$

and then

$$\mathbf{d} = \frac{\sum_{i=1}^{K} v_i \mathbf{d}'_i}{\delta + \sum_{i=1}^{K} v_i}, \mathbf{f} = \frac{\sum_{i=1}^{K} v_i w_i \mathbf{f}_i}{\delta + \sum_{i=1}^{K} v_i}, \mathbf{m} = \frac{\sum_{i=1}^{K} v_i w_i \mathbf{m}_i}{\delta + \sum_{i=1}^{K} v_i}, \tag{7}$$

where $H$ is the unit step function and $\delta$ is a small value to avoid numeric errors. $v_i = 0$ when $\mathbf{x}$ is not observable in camera $i$, because if $\mathbf{x}$ is occluded in camera $i$, it should not contribute to the descriptor of $\mathbf{x}$. In addition, we could only have a confident estimation when $\mathbf{x}$ is close to the surface. Therefore, $w_i$ will decay as $|\mathbf{d}_i|$ increases. For $\mathbf{x}$ that is far away, $\mathbf{f}$ and $\mathbf{m}$ will degrade to $0^T$.

We convert the implicit field function $\mathcal{F}(\cdot)$ to a set of keypoints $\boldsymbol{s}$. First, we create voxels $\mathbf{x} \in \mathbb{R}^{W \times L \times H \times 3}$ in the workspace and evaluate $(\mathbf{d}, \mathbf{f}, \mathbf{p}) = \mathcal{F}(\mathbf{x})$. We filter out $\mathbf{x}_i \in \mathbf{x}$ where $\mathbf{d}_i$ is large or $\mathbf{p}_i$ has a low probability to avoid empty space and the background. After obtaining filtered points $\mathbf{x}'$, we use farthest point sampling to find surface points $\boldsymbol{s} \in \mathbb{R}^{3 \times n_s}$ of an instance.

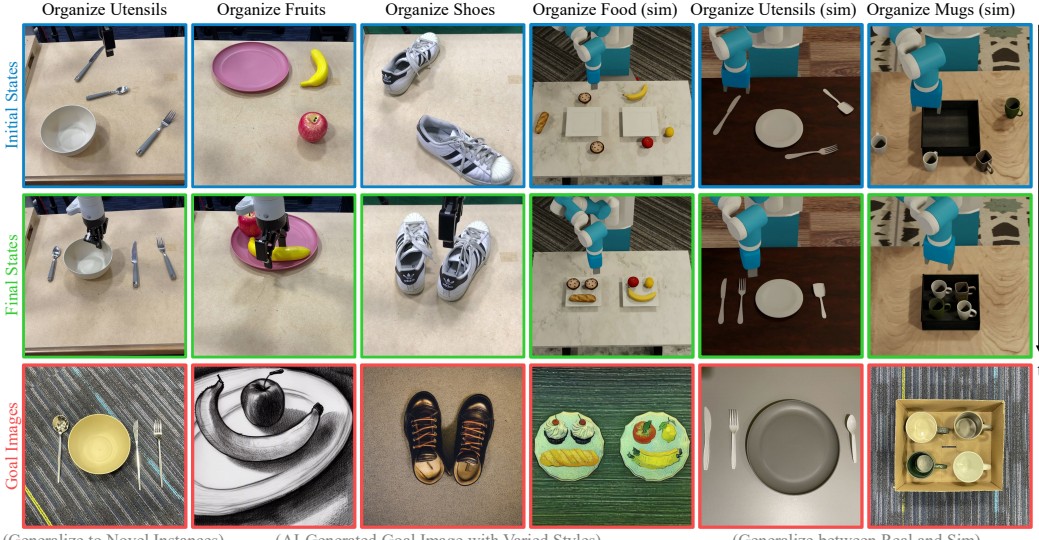

Figure 4: **Qualitative results.** We qualitatively evaluate our proposed framework on household manipulation tasks, both in the real world and in simulation, encompassing tasks such as organizing utensils, fruits, shoes, food, and mugs. The figure highlights that our representation can generalize across varied instances, styles, and contexts. For instance, in the organizing fruits example, the goal image, unlike the workspace, is styled as a sketch drawing. Because our representation can map bananas with varied styles and appearances to similar features, the banana in the workspace can correspond to the banana in the sketch. This allows the task to be successfully completed. This wide range of tasks showcases the generalization capabilities and manipulation precision of our framework.

## 3.4 Keypoints Tracking and Dynamics Training

This section will present how to use the dynamic implicit 3D descriptor field $\mathcal{F}(\cdot)$ to track keypoints and train dynamics. Without losing generalization, consider the tracking of a single instance $s^t \in \mathbb{R}^{3 \times n_s}$. For clarity, we denote $\mathbf{f}$ and $\mathbf{d}$ from $\mathcal{F}(\cdot)$ as $\mathcal{F}_{\mathbf{f}}(\cdot)$ and $\mathcal{F}_{\mathbf{d}}(\cdot)$. We formulate the tracking problem as an optimization problem:

$$\min_{s^{t+1}} \quad ||\mathcal{F}_{\mathbf{f}}(s^{t+1}) - \mathcal{F}_{\mathbf{f}}(s^0)||_2. \tag{8}$$

Since $\mathcal{F}(\cdot)$ is differentiable, we could use a gradient-based optimizer. This method could be naturally extended to multiple-instance scenarios. We found that relying solely on features for tracking is unstable. We added rigid constraints and distance regularization for a more stable tracking.

Keypoint tracking enables dynamics model training on real data. We instantiate the dynamics model $f(\cdot, \cdot)$ as graph neural networks (GNNs). We follow [59] to predict object dynamics. Please refer to [25, 59] for more details on how to train the GNN-based dynamics model. The trained dynamics will be used for trajectory optimization in Section 3.5.

## 3.5 Zero-Shot Generalizable Robotic Manipulation

As described in Section 3.3, we denote initial tracked points and features as $s^0$ and $\mathbf{f}^0$. We estimate $s_{\text{goal}} \in \mathbb{R}^{2 \times n_s}$ of goal image $\mathcal{I}_{\text{goal}}$ as follows:

$$\alpha_{ij} = \exp\left(||\mathcal{W}^{\mathbf{f}}_{\text{goal}}[\mathbf{u}_i] - \mathbf{f}^0_j||_2\right),$$
$$w_{ij} = \frac{\exp\left(s\alpha_{ij}\right)}{\sum_{i=1}^{H \times W} \exp\left(s\alpha_{ij}\right)}, \tag{9}$$

then we have $s_{\text{goal},j} = \sum_{i=1}^{H \times W} w_{ij} \mathbf{u}_i$, where $\mathcal{W}^{\mathbf{f}}_{\text{goal}}$ is the feature volume extracted from $\mathcal{I}_{\text{goal}}$ using DINOv2. $s$ is the hyperparameter to determine whether the heatmap $w_{ij}$ is more smooth or concentrating. Although Eq. 9 only shows a single instance case, it could be naturally extended to multiple instances by using instance mask information.

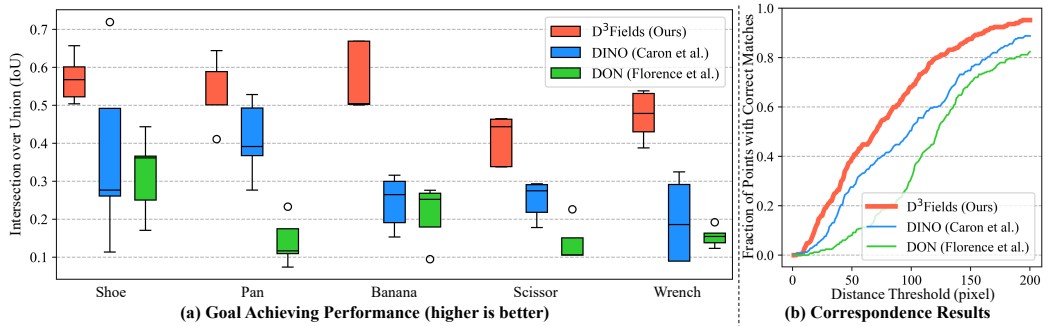

**(a) Goal Achieving Performance (higher is better)**    **(b) Correspondence Results**

Figure 5: **Quantitative Evaluation.** We perform real-world quantitative evaluations by measuring final goal-achieving performance and keypoints correspondence accuracy. (a) We use IoU to measure goal-achieving performance. Results indicate that our method aligns with the goal configurations much better than DON and DINO across various object categories and scenarios. (b) We measure the keypoints correspondence accuracy according to the fraction of points with accurate matches, with correct matches determined by a distance threshold. Our method is consistently better at aligning with the goal image, regardless of the chosen threshold.

However, $s_{\text{goal}}$ is in the image space, while $s^t$ is in the 3D space. We bridge this gap by introducing a reference camera with approximate intrinsic and extrinsic parameters $\mathbf{K}'$ and $\mathbf{T}'$. Instead of rendering images in the reference view, we focus on projecting 3D keypoints into 2D images and define the task cost function in image space as follows:

$$c(s^t, s_{\text{goal}}) = ||\pi \left( \mathbf{K}' \left( \mathbf{R}' s^t + \mathbf{t}' \right) \right) - s_{\text{goal}}||_2^2. \tag{10}$$

## 4   Experiments

In this section, we evaluate our representation across various manipulation tasks with varying goal image styles, instances, and contexts. We visualize D$^3$Fields and showcase tracking results in Section 4.2. Then, we highlight our framework's zero-shot generalizability in both real-world and simulated tasks in Section 4.3. Finally, a quantitative comparison with baselines in Section 4.4 underscores our framework's generalization and manipulation precision.

### 4.1   Experiment Setup

In the real world, we employ four OAK-PRO D cameras to gather RGBD observations and use the Kinova® Gen3 for action execution. In simulation, we utilize OmniGibson and deploy Fetch for mobile manipulation tasks [60]. Our evaluations span a variety of tasks, including organizing shoes, collecting debris, tidying the office table, arranging utensils, and more.

We implement the baseline methods using Dense Object Nets (DON) and DINO for feature extraction [61, 54]. We quantitatively evaluate these methods on five object classes for single-instance manipulation tasks in the real world. The results and analysis are presented in Section 4.4.

### 4.2   Descriptor Fields Visualization and Keypoints Tracking

D$^3$Fields provide a good 3D semantic representation, as shown in Fig. 3(a). We first visualize the mask fields by coloring 3D points according to their most likely instance, and our visualization shows a clear 3D instance segmentation. Additionally, we map the semantic features to RGB space using PCA, as with DINOv2 [17]. Visualization of the descriptor fields reveals that D$^3$Fields retain a dense semantic understanding of objects. In the provided shoe example, even though various shoes have distinct appearances and poses, they exhibit similar color patterns: shoe heels are represented in green, and shoe toes in red. We observed similar patterns when evaluating the model on mugs and forks.

As discussed before, D$^3$Fields can also capture scene dynamics. We evaluate it by tracking the object keypoints. We show two examples of 3D keypoint tracking in Fig. 3(b). In the first example,

a shoe is pushed and then flipped. Although only a portion of the shoe is visible from the view, our framework tracks it reliably. In another example, a shoe is lifted and then set down. Despite parts of the shoe being out of the camera's view, we can robustly track it in 3D.

### 4.3 Zero-Shot Generalizable Manipulation

We conduct a qualitative evaluation of $D^3$Fields in common household robotic manipulation tasks in a zero-shot manner, with partial results displayed in Fig. 1 and Fig. 4. The following capabilities of our framework are observed:

**Generalization to AI-Generated Goal Images.** In Fig. 1, the goal image, rendered in a Van Gogh style, depicts shoes distinct from those in the workspace. Since $D^3$Fields encode semantic information, capturing shoes with varied appearances under similar descriptors, our framework can manipulate shoes based on AI-generated goal images.

**Compositional Goal Images and 3D Manipulation.** Using the office desk organization example in Fig. 1, the robot first arranges the mouse and pen according to the goal image. It then repositions the mug from the box to the mug pad, referencing a goal image of the upright mug.

**Generalization across Instances and Materials.** Granular objects, unlike rigid ones, have more complex dynamics. Our framework effectively handles these materials, as shown in the debris collection in Fig. 1. Fig. 4 further showcases our framework's instance-level generalization, where the goal image displays instances different from the workspace.

**Generalization across Simulation and Real World.** We evaluated our framework on household tasks in the simulator, as shown in the utensil organization and mug organization examples in Fig. 4. Given goal images taken from the real world, our framework can also manipulate objects to the goal configurations. Our framework demonstrates generalization capabilities between simulation and the real world.

### 4.4 Quantitative Comparisons with Baselines

In Fig. 5(a), we measure performance using the IoU between the goal image mask and the final state mask after manipulation, with higher values indicating better alignment. Evaluating across five object classes, our method consistently outperforms the baselines, underscoring its generalization and manipulation accuracy. While DINO struggles with distinguishing object components, leading to imprecise results, it still works better than DON. Although DON performs well on familiar object classes and configurations, it lacks generalization in novel scenarios.

In Fig. 5(b), we present the correspondence results. We manually label corresponding keypoints on both the goal image and the final manipulation result to evaluate the correspondence accuracy. We calculate the fraction of accurately matched points based on a distance threshold. Our method consistently outperforms the baselines, regardless of the threshold. DINO ranks second, while DON lags behind. Consistent with Fig. 5(a), our method excels in generalization and accuracy, DINO is broadly applicable but less precise, and DON struggles with generalization.

## 5 Conclusion

In this work, we introduce $D^3$Fields, which implicitly encode 3D semantic features and 3D instance masks, and model the underlying dynamics. Our emphasis is on zero-shot generalizable robotic manipulation tasks specified by 2D goal images of varying styles, contexts, and instances. Our framework excels in executing a diverse array of household manipulation tasks in both simulated and real-world scenarios. Its performance greatly surpasses baseline methods such as Dense Object Nets and DINO in terms of generalization capabilities and manipulation accuracy.

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
