# OpenReview forum: "D$^3$Fields: Dynamic 3D Descriptor Fields for Zero-Shot Generalizable Robotic Manipulation"
_robot-learning.org/CoRL/2023/Workshop/TGR — CoRL 2023 Workshop TGR Poster_

### Official Review · Reviewer_vg6W · 2023-10-16

**Rating:** 7
**Confidence:** 3

**Review:**

This paper proposes a representation to describe 3D, dynamic, and semantic information. Experiments show that this representation is effective for manipulation tasks in both simulation and real-world environment. This could be a promising building block for generalist robots' perception and reasoning modules.

---

### Decision · Program_Chairs · 2023-10-21

**Decision:**

Accept (Poster)

**Comment:**

Great paper!